# *TP53* Gene Therapy as a Potential Treatment for Patients with COVID-19

**DOI:** 10.3390/v14040739

**Published:** 2022-03-31

**Authors:** Joe B. Harford, Sang Soo Kim, Kathleen F. Pirollo, Esther H. Chang

**Affiliations:** 1SynerGene Therapeutics, Inc., Potomac, MD 20854, USA; kimss@synergeneus.com; 2Department of Oncology, Georgetown University Medical Center, Washington, DC 20007, USA; pirollok@georgetown.edu (K.F.P.); change@georgetown.edu (E.H.C.)

**Keywords:** SARS-CoV-2, COVID-19, p53, TP53, gene therapy, innate immunity, host antiviral defense

## Abstract

SGT-53 is a novel investigational agent that comprises an immunoliposome carrying a plasmid vector driving expression of the human *TP53* gene that encodes wild-type human p53. SGT-53 is currently in phase II human trials for advanced pancreatic cancer. Although p53 is best known as a tumor suppressor, its participation in both innate and adaptive immune responses is well documented. It is now clear that p53 is an important component of the host response to various viral infections. To facilitate their viral life cycles, viruses have developed a diverse repertoire of strategies for counteracting the antiviral activities of host immune system by manipulating p53-dependent pathways in host cells. Coronaviruses reduce endogenous p53 levels in the cells they infect by enhancing the degradation of p53 in proteasomes. Thus, interference with p53 function is an important component in viral pathogenesis. Transfection of cells by SGT-53 has been shown to transiently produce exogenous p53 that is active as a pleiotropic transcription factor. We herein summarize the rationale for repurposing SGT-53 as a therapy for infection by SARS-CoV-2, the pathogen responsible for the COVID-19 pandemic. Because p53 regulation was found to play a crucial role in different infection stages of a wide variety of viruses, it is rational to believe that restoring p53 function based on SGT-53 treatment may lead to beneficial therapeutic outcomes for infectious disease at large including heretofore unknown viral pathogens that may emerge in the future.

## 1. Introduction

Severe acute respiratory syndrome coronavirus 2 (SARS-CoV-2) is the causative agent responsible for the COVID-19 pandemic [1,2,3]. SARS-CoV-2 shares substantial nucleotide homology with SARS-CoV-1 and Middle East respiratory syndrome coronavirus (MERS-CoV), coronaviruses that are also capable of causing life-threatening human respiratory diseases [4]. Although effective vaccines against SARS-CoV-2 have been developed and deployed, substantial numbers of new infections continue to occur worldwide. Even in the U.S., where vaccines are available, coverage varies by locale, and those under 5 years old are not currently eligible for vaccination. Genetic variants of SARS-CoV-2 (e.g., the delta variant, the omicron variant, etc.) that are more transmissible have arisen, and concerns have been raised about variants arising that are capable of evading the immunity afforded by prior infection or currently available vaccines. Accordingly, there remains an urgent need for antiviral therapeutics with efficacy against multiple variants of SARS-CoV-2. During evolutionary adaptation to their hosts, many viruses including coronaviruses have developed a range of strategies to manipulate host p53 either by reducing its level or altering its activity [5,6,7,8,9,10]. In the case of coronaviruses, the levels of endogenous p53 are reduced by a protease with deubiquitinating activity encoded by the virus [11,12]. Here, we set forth the rationale for assessing *TP53* gene therapy as an antiviral countermeasure against SARS-CoV-2 infections taking advantage of an investigational agent originally designed for use in oncology.

## 2. The Role of p53 in Immunity

The p53 protein, encoded by the *TP53* gene, is among the most studied of molecules with more than 80,000 publications arising from nearly 40 years of research [9,13]. P53 has been called “guardian of the genome” [14] based on its role in responding to DNA damage via triggering cell cycle arrest, apoptosis, and/or senescence (the “canonical functions” of p53). The vast majority of studies involving p53 focus on oncology, reflecting the fact that *TP53* is the most commonly mutated gene in human cancers. More than 50% of human tumors carry *TP53* mutations, although the prevalence of p53 mutations differs depending on the tumor’s site of origin [15,16]. Although better known as a tumor suppressor, the involvement of p53 in the regulation of immune responses is also well documented [9,17,18,19]. A linkage between p53 and the immune system was evident in very early studies with p53 null (p53^−/−^) mice. These mice lacking the *TP53* gene developed tumors as anticipated, but approximately one-quarter of the mice died of unresolved infections prior to the appearance of tumors, suggesting that p53^−/−^ mice have a severely compromised immune system [20]. Subsequently, p53^−/−^ mice were shown to exhibit more severe disease compared to their p53^+/+^ counterparts after infection by influenza A virus, another respiratory virus [21,22]. Analyses of the defective antiviral response in the p53^−/−^ mice implicated p53 as a component of both innate and adaptive antiviral immunity. Based on its role in immunity, it has been suggested that the title “guardian of immune integrity” be added to the more familiar title “guardian of the genome” [18]. P53 is a pleiotropic transcription factor with binding sites for p53 existing in the promoter regions of literally hundreds of genes [13,23]. It is, therefore, perhaps not surprising that regulation by p53 has been implicated for a number of genes involved in immune signaling, autoimmunity, post-apoptotic dead cell clearance, immune tolerance, and immune checkpoint regulation [9,18].

## 3. Evidence for p53 Involvement in Broadly Opposing Infections by Viruses

Collectively, viruses have acquired an impressive repertoire of mechanisms for manipulating the p53 of their host [5,6,7,8,9,10]. Some viruses counter p53′s antiviral effects by the deployment of distinct viral proteins that subvert p53′s functions. Indeed, p53 was originally identified in association with SV40 polyomavirus large T antigen [9] that binds to the region of p53 that interacts with DNA and, thereby, blocks p53-specific transcriptional control [24]. Adenoviruses also encode proteins that interact with the p53 DNA-binding region to block its role as a transcription factor [25]. The NS1 protein of influenza A virus also interacts with p53 and alters its binding to the promoters of p53-responsive genes [26]. In contrast, other viruses reduce p53 levels by destabilizing p53 using the host’s machinery for protein turnover or by producing virally encoded proteases capable of degrading p53 [8]. Human papilloma virus (HPV) is an example of a virus that modifies the stability of p53. The HPV E6 protein interacts with an E3 ubiquitin ligase (E6-AP) leading to ubiquitination of p53 and its degradation in proteasomes [27,28]. The result is a reduction in p53 levels in HPV-infected cells, thereby creating a more HPV-friendly environment. The variety of ways for viruses to thwart the antiviral activity of p53 suggests that these mechanisms were acquired independently during the course of evolution and speaks to the importance of p53 as a critical component of the host antiviral defense (Figure 1). However, despite great potential, p53 has not received much attention in the context of antiviral therapies.

Triggering apoptosis in response to DNA damage is a “canonical” function of p53 that is central to its role as “guardian of the genome” [14]. Apoptosis mediated by p53 also occurs in response to foreign DNA or RNA brought into cells as the genome of an invading viral pathogen [29]. One can consider the programmed death of a virally infected cell in terms of “altruistic suicide” [30]. Cellular apoptosis at an early stage of infection would be detrimental to viral replication and spread if individual infected host cells die before infectious viral progeny are produced to infect neighboring cells. Thus, many viruses seek to block the pro-apoptotic activity of p53. For example, the hepatitis B virus encodes X protein that binds to p53 and blocks its DNA binding to abrogate apoptosis [31,32]. This decrease in apoptosis appears to be mediated through the *BBC3* gene encoding “p53 upregulated modulator of apoptosis” (PUMA), a pro-apoptotic protein induced by p53 [33]. The *BAX* gene that encodes a downstream regulator of apoptosis whose expression is directly upregulated by p53 at the transcriptional level also appears to be involved [34]. On the other hand, certain viruses trigger apoptosis as a part of their life cycle. The induction of apoptotic cell death is a hallmark of influenza infection, and p53 is essential for this to occur [35]. In mouse embryo fibroblasts isolated from p53^−/−^ mice, influenza-virus-induced apoptosis was absent. After infection, viral titers were higher in the fibroblasts from the p53-knockout mice. These authors postulated that viral titers being higher when p53 levels were lower was due to a downmodulation of the interferon (IFN) response in the infected cells.

In addition to its role in apoptosis, the antiviral activity of p53 involves orchestrating diverse signaling pathways originating from many different cellular receptors and sensors including type 1 IFN (IFN-1). As a pleiotropic transcription factor, p53 regulates the expression of key components of the host’s antiviral response including interferon regulatory factors (IRFs), protein kinase RNA-activated (PKR), toll-like receptor 3 (TLR3), IFN-stimulated gene 15 (ISG15), and monocyte chemoattractant protein 1 (MCP1 also known as CCL2) [10]. IFN-1 is a well-studied component of the body’s antiviral response, and regulation of its activity is complicated involving several p53-regulated cellular pathways. The expression of p53 itself is induced after IFN treatment [36]. Thus, IFN regulates p53 expression, and p53 in turn regulates IFN production, suggesting that exists an IFN–p53 positive feedback loop to amplify the cellular response to viral infections. The expression of IFN-1 is under the control of a number of regulators including interferon regulatory factors (e.g., IRF9 and IRF7). IRF9 is directly regulated transcriptionally by p53 [37] as is IRF7 [11]. When p53 levels are high, IRF expression is high, and therefore, IFN-1 production is elevated. If a virus can somehow disable p53, it would, thereby, reduce key IRF levels and consequently reduce IFN-1 production. The disabling of p53 by one means or another would, thereby, allow a virus to evade the antiviral effects of IFN-1. IFN signaling and the induction of apoptosis are interconnected. IFN signaling drives increased p53 mRNA and protein levels in order to evoke more-robust p53 responses that trigger apoptosis of infected cells and restrict virus replication.

## 4. Evidence for p53 Involvement in Combatting Infections by Coronaviruses

Although the details of how SARS-CoV-2 manipulates the p53 pathway are just beginning to emerge, more detailed information is available on other members of the coronavirus family including SARS-CoV-1, which was responsible for the SARS outbreak of 2002–2004 and MERS-CoV, responsible for the outbreak in the Middle East in 2012 [4]. The COVID-19 pandemic has far surpassed both these previous human coronavirus outbreaks in terms of worldwide infections, hospitalizations, deaths, and economic impact. The signaling pathways involved in infection and pathogenesis of SARS-CoV-1 and MERS-CoV are better understood, and it has been hypothesized that these pathways are shared by the closely related SARS-CoV-2 [38]. The involvement of the IFN-1 response in opposing SARS-CoV-2 infection is also likely to be similar to that observed with SARS-CoV-1 and MERS-CoV [39]. Use of IFN-1 against COVID-19 has been proposed [40], and administration of IFNα by vapor inhalation is included in Chinese guidelines for the treatment of patients with COVID-19 [41].

The host immune system undergoes profound and complex changes during symptomatic COVID-19 disease, and a number of cell types participate in the host response [42]. In patients with COVID-19, the overactivation of the inflammatory immune response can lead to a cytokine storm and immune exhaustion. Based on infection of primates with SARS-CoV-1, it appears that expression of the anti-inflammatory activity of IFN-1 can prevent tissue injury [43]. Interestingly, older animals infected with SARS-CoV-1 expressed lower levels of IFN-1 than younger animals, and in the COVID-19 pandemic, older individuals are at higher risk of serious illness and death.

SARS-CoV-2 also restrains antigen presentation by downregulating the major histocompatibility complex (MHC) class I and II proteins, thereby curtailing adaptive immune responses mediated by T cells [44]. Macrophages and dendritic cells, the primary antigen-presenting cells, are found in all human organs, but for SARS-CoV-2 and other respiratory viruses, a first line of defense is thought to be lung alveolar macrophages (AMs) [45,46]. AMs are known to be one of the main producers of IFN-1s during infection by respiratory viruses such as influenza virus [47,48] or respiratory syncytial virus [49]. AMs have on their surface angiotensin-converting enzyme 2 (ACE2), which serves as the receptor for SARS-CoV-2 [50]. It has been suggested that, in addition to ACE2, SARS-CoV-2 can also use other cellular proteins as receptors. On certain immune cells, CD147 and/or CD26 appear to serve as SARS-CoV-2 receptors [51]. Compared to other respiratory viruses, coronaviruses are poor inducers of IFN-1, having the ability to escape and counteract innate sensing and IFN production [39,52,53]. It has been reported that AMs infected with SARS-CoV-2 do not produce a robust IFN response, and absence of IFN production in AMs appears to contribute to the severity of SARS-CoV-2 infections [54]. The inhibition of a robust IFN-1 response has also been associated with clinical severity after infection with SARS-CoV-1 or MERS-CoV [55]. Key players in the production of IFN-1 response are plasmacytoid dendritic cells (pDCs), which produce 1000-fold more IFN-1 than any other cell type [56,57,58] and are seemingly crucial in controlling coronavirus infections [59,60]. These cells express high levels of the toll-like receptors (TLRs) used to recognize the nucleic acids derived from viral pathogens, and they bridge innate and adaptive immune responses by not only producing IFN-1 but also serving as antigen-presenting cells.

A powerful case for the importance of p53 in preventing infection by coronaviruses is made by the fact that these viruses, like many other viruses, have evolved specific means to oppose p53 functions by reducing p53 levels [10]. It has been inferred that p53 is a coronavirus antagonist based on finding that nonstructural proteins encoded by coronaviruses result in the destruction of endogenous p53 and that p53 expression inhibits replication of infectious SARS-CoV-1 as well as human coronavirus NL63 [28]. Based on these findings, it was concluded that p53 is “a major player in antiviral innate immunity” against coronaviruses.

A detailed description of the replication and pathogenesis of coronaviruses is beyond the scope of this article. Suffice to say here that the primary means by which coronaviruses combat the p53-dependent antiviral activities appears to be proteolytic destruction of endogenous p53. It is of note that the destabilization of p53 is one of the very commonly employed strategies used by viruses seeking to reduce p53 levels to their own benefit [1]. The coronavirus genome encodes polyproteins that are processed by virally encoded proteases to generate 16 nonstructural proteins [61]. Polyprotein processing involves virally encoded enzymes that are members of the papain family of cysteine proteases (termed PLP or PL^pro^ depending on the coronaviruses). In addition to one or more PLPs, a 3C-like protease is also encoded by coronaviruses. The coronaviral PLP domain that is within the nonstructural protein 3 (nsp3) region of its genome also has a deubiquitinating activity [62] that is capable of deubiquitinating and, thereby, stabilizing the cellular E3 ubiquitin ligases RCHY1 and/or MDM2 [11,12]. MDM2 is known to interact with and ubiquitinate p53 leading to its degradation in proteasomes [63]. When more MDM2 (or RCHY1) is present, p53 is more extensively ubiquitinated and degraded [64,65]. The principal mechanism by which p53 levels are kept at low levels in uninfected cells involves its ubiquitination and degradation in proteasomes [66]. Irrespective of whether p53 destruction in coronavirus-infected cells involves only host proteases or uses one or more of the viral proteases, the effect of deubiquitination of the host’s E3 ubiquitin ligases by virally encoded nsp3 is a decrease in cellular p53 levels. This reduction in p53 leads to the reduced production of IRFs that are directly transcriptionally regulated by p53, and so less IFN-1 is produced. Reduction of p53 levels would also be expected to negatively impact the induction of apoptosis in response to coronavirus-derived nucleic acids and the expression of the genes of innate and adaptive immunity that are regulated by p53. Thus, the coronaviruses create a more “virus friendly” cell through virally encoded proteins that act to destabilize p53. Nutlins are small-molecule antagonists of MDM2 that interfere with the MDM2–p53 interaction resulting in the stabilization of p53 [67]. It has been proposed by others that a member of the Nutlin family (idasanutlin) might have anti-coronavirus activity via the stabilization of p53 [68].

Insights into how the manipulation of p53 levels by coronaviruses might be exploited therapeutically come from a serious and highly contagious disease affecting young pigs. Porcine epidemic diarrhea is caused by a coronavirus (PEDV) that is in the same subfamily as SARS-CoV-1, MERS-CoV, and SARS-CoV-2 [69]. PEDV manipulates host immunity to benefit its own replication, including cell cycle arrest [70], which is one of the canonical activities of p53. PEDV, similar to its coronavirus relatives, encodes a PLP and negatively affects the production of IFN-1 [71]. It has been shown that p53 levels rise as part of the antiviral response after PEDV infection and that loss of p53 promotes PEDV replication [72]. From the perspective of therapeutic strategies, the observation that treatment with a known activator of p53 (Nutlin-3) or upregulation of p53 inhibited PEDV infection is highly relevant. Here, we propose to upregulate p53 levels via *TP53* gene therapy.

## 5. *TP53* Gene Therapy as an Antiviral Strategy

Given that p53 has been demonstrated to be a component of the body’s antiviral response, the question is, would more p53 make cells more resistant to viral infection? A similar question arose with regard to cancer, i.e., Would more p53 make animals more resistant to tumors? A direct correlation between p53 levels and resistance to tumor development has been addressed by generating mice with an extra copy of p53 (“super p53” mice) [73]. “Super p53” mice exhibit enhanced DNA damage response, are tumor resistant, and age normally. Subsequently, it was shown that vesicular stomatitis virus replication in mouse embryo fibroblasts derived from “super p53” mice is impaired as a result of enhanced apoptosis induced by p53 activation [21]. These findings suggest that having extra copies of the *TP53* gene or otherwise increasing p53 levels makes cells more resistant to viral infection.

We are developing a nanomedicine termed SGT-53 (also called scL-p53) for *TP53* gene therapy in oncology that employs a delivery system termed scL (for single-chain liposome). SGT-53 comprises a cationic liposome that is decorated on its surface with a single-chain antibody fragment directed against the human transferrin receptor (TfR) and carrying a plasmid designed to express the gene encoding human p53 (Figure 2A). The elevated levels of TfR (also known as CD71) on tumor cells have long been recognized as a means of targeting therapeutics to cancer cells [74]. The endothelial cells that form the blood–brain barrier use TfR-mediated transcytosis to move diferric transferrin from the blood into the brain, so the TfR can be used to ferry therapeutics across the blood–brain barrier [75,76,77]. Consequently, when SGT-53 is administered intravenously to mice bearing intracranial glioblastoma tumors, the nanocomplex crosses the blood–brain barrier and is taken up by the tumors [78]. Once in the intracranial tumors, the *TP53* gene payload of SGT-53 is expressed as active p53 protein that affects the expression of genes downstream of p53 in cellular pathways (Figure 2B) [79]. MGMT is involved in DNA repair, p21 participates in cell cycle control and cPARP is upregulated in apoptotic cell death. After treatment with SGT-53, we have observed an increase in apoptosis in human tumor xenografts [80] and in several syngeneic murine models for various tumor types [81,82,83].

From the perspective of p53’s role in opposing viral infections, the recovery of p53 function in infected cells could result in elimination of the infected cells before viral replication takes place, thus, vastly limiting the magnitude of the infection. It is also clear that IFN-1 production is a key element in the antiviral response. When mouse cells were transfected with SGT-53, the levels of expression of IFN-1 genes (*Ifna2* and *Ifnb2*) were markedly upregulated in a p53-specific manner, i.e., the nanocomplex carrying plasmid vector lacking the *TP53* gene (scL-Vec) did not increase IFN-1 gene expression (Figure 3). SGT-53 has been shown to be well tolerated with some indications of anticancer activity in phase Ia and Ib trials [84,85] and is now being tested against advanced pancreatic cancer in combination with gemcitabine/nab-paclitaxel in a phase II trial (ClinicalTrials.gov Identifier: NCT02340117). SGT-53 is among a small number of receptor-targeted nanomedicines now undergoing clinical evaluation [86,87].

Based on the growing recognition that p53 regulates immune responses, we have studied SGT-53 for its ability to augment cancer immunotherapy in a number of syngeneic mouse models including tumors of the head and neck, glioblastoma and breast, lung, and kidney cancers [81,82,83]. The findings of all these studies indicate that increasing levels of p53 in tumors via intravenous administration of SGT-53 makes immunotherapy based on checkpoint inhibitors more effective. Extensive analyses of transcripts in tumor tissues are consistent with p53 acting to alter expression of a number of genes known to be involved in innate and adaptive immunity. Immunosuppression was reduced by SGT-53 treatment, and tumors appeared to be more immunologically “hot”. Using a NanoString^®^ panel of 225 genes of relevance to innate immunity, we found 26 genes to be significantly upregulated in tumor tissue when mice bearing syngeneic LL/2 lung tumors were treated with SGT-53. These data are consistent with p53 being a pleiotropic transcription factor that can regulate innate immunity either directly or indirectly. In addition, SGT-53 treatment led to higher levels of transcripts known to be expressed in macrophages of the classically activated M1 class and lower levels of mRNAs found in alternatively activated M2 macrophages. Participation of M1 macrophages is considered to be antitumoral, and M1 macrophages also participate in early antiviral responses [88].

The cells of the airway epithelium have long been recognized as the point of entry for inhaled respiratory viruses including SARS-CoV-2 [89,90]. Airway epithelial cells also represent a first line of defense against these viruses via their mediation of host innate immunity. From the perspective of using SGT-53 as an antiviral agent against SARS-CoV-2 (or future emerging respiratory viral pathogens), the objective would be to curtail viral replication and spread from the initially infected cells. To accomplish this goal, it would be important that p53 from our nanocomplexes is expressed in cells where the viruses replicate, i.e., in the airway epithelium. SGT-53 enters cells via the recognition of TfRs by nanocomplex’s targeting moiety, and active p53 is expressed in cells after internalization (see Figure 2). Lung epithelial cells clearly express TfRs as evidenced by the finding that certain New World arenaviruses actually utilize TfRs as their cellular receptor for infection of airway epithelial cells [91,92]. Airway epithelial cells should therefore be targeted by SGT-53.

As described above, coronaviruses, including SARS-CoV-2, do not appear to elicit as much IFN-1 as do other viruses. One possible reason for the absence of an adequate IFN response may be reduction of endogenous p53 as a result of its proteolysis by the coronaviral PLP. Boosting p53 levels using SGT-53 in AMs is certainly feasible since in addition to ACE2 on their plasma membranes, AMs also express TfR. As monocytes mature to AMs, expression of TfR increases, and it has been proposed that TfR be regarded as a differentiation-dependent marker of AMs [93,94]. This means that AMs would be targeted by the targeting moiety of SGT-53, i.e., the single-chain antibody fragment recognizing TfR. Another important cell type in host antiviral response at an early stage of infection is pDCs, which serve as a primary source of IFN-1. It is known that pDCs also have TfR on their surface [95], making them too a target for SGT-53-mediated *TP53* gene therapy.

## 6. Discussion

We hypothesize that a battle between SARS-CoV-2 and its host takes place around p53 (Figure 4). Similar to many other types of viruses, the coronaviruses aim to counter p53-mediated antiviral responses that hamper viral replication and spread. The fact that mice lacking p53 are more permissive of coronavirus infection is strong evidence that p53 acts to inhibit these viruses [12]. Mice lacking p53 display a more severe disease induced by another respiratory virus (influenza A virus) compared to their p53^+/+^ counterparts [21], and these mice displayed impairment of both their innate and adaptive antiviral immunity. A complimentary result was seen in pig cells; wherein, cells in which p53 was knocked out were more productively infected with PEDV, i.e., more viral progeny were seen with p53-knockout pig cells than with p53-wild-type cells [72].

Can increasing p53 levels via *TP53* gene therapy become a therapeutic option to limit the viral replication or dissemination of coronaviruses? Available data would argue in the affirmative. Even a modest increase in p53 levels seen in the so-called “super p53” mice with an extra copy of the mouse p53 gene were more resistant to vesicular stomatitis virus [21]. We have shown that the nanocomplex termed SGT-53 is capable of pushing p53 levels up substantially and that the exogenous p53 produced is fully capable of driving the expression of genes known to be regulated by p53 including IFN-1 genes. We envision that coronaviruses destabilize and degrade endogenous p53 in the cells they infect, resulting in an IFN response that is less than robust and ineffective for the inhibition of viral replication and spread. By increasing the cellular level of p53 using SGT-53, we propose that the ability of the coronaviruses to replicate would be curtailed. During their evolution, virtually all viruses have developed means to counter host defense mechanisms that aim to hamper viral replication and spread. Because p53 is an important component of host immunity, p53 represents a threat that viruses seek to neutralize or circumvent in some way. Dr. Arnold Levine, one of the discoverers of p53, has recently gone so far as to state that “almost every successful virus has developed ways to inactivate p53” [9]. Sadly, SARS-CoV-2 has proven to be a very successful virus. Despite progress on the development and distribution of vaccines against SARS-CoV-2, new infections, hospitalizations, and deaths continue to mount worldwide and will likely continue for the foreseeable future. As of 13 March 2022, confirmed global infections totaled nearly 500 million and global deaths had exceeded 6 million [96]. Thus, the development of effective treatment for individuals infected with SARS-CoV-2 remains an urgent unmet medical need. SGT-53 was designed as an oncology product for expressing the normal human p53 tumor suppressor in cancer cells to sensitize these cells to chemotherapy and radiotherapy [97]. One advantage of repurposing this investigational agent as an antiviral countermeasure is that SGT-53 has already completed successful phase Ia and Ib human trials with a good safety profile and is now in a phase II clinical trial. Since SGT-53 has already been produced as a clinical-grade (cGMP) product and assessed for safety, the ability to conduct SGT-53 clinical trials in patients with COVID-19 will be expedited compared to the development of an entirely new therapeutic agent.

It is anticipated that proof-of-principle experiments assessing *TP53* gene therapy as an antiviral approach might involve the use of cell culture and animal models for SARS-CoV-2 infection. Critical to these experiments would be examination of the timing of SGT-53 dose(s) relative to viral exposure and the appearance of the symptoms of infection. Although using gene therapy for a viral infection may seem radical to some, it is noteworthy that the *TP53* gene carried by SGT-53 does not incorporate into the genome but is expressed transiently with exogenous p53 detectable in transfected cells within 6 h after intravenous administration, with peak expression of exogenous p53 at 24–48 h and markedly diminished p53 expression at 72 h [93]. Moreover, the exogenous *TP53* gene being delivered by SGT-53 is a gene that already exists in all normal human cells. In both adult and pediatric patients with cancer receiving SGT-53, we have not observed side effects related to potentially increased expression of p53 in normal cells. Interestingly, elephants have up to 20 copies (40 alleles) of the *TP53* gene rendering the elephant’s cells more prone to apoptosis in response to DNA damage but not triggering apoptosis in undamaged normal cells [98].

Given that p53 is a transcription factor with many regulated genes under its control, the introduction of an exogenous *TP53* gene would ripple into the various cellular pathways. Once expressed, the exogenous p53 would, among other things, be expected to restore apoptosis in virally infected cells and to drive production of IFN-1 and a number of other genes involved in innate and adaptive immune response to the invading pathogen. In fact, p53 is a very pleiotropic transcription factor with a far-reaching impact on cellular processes, e.g., ionizing radiation that creates DNA damage that triggers the p53-dependent upregulation of about 500 genes coupled with the downregulation of a lesser number [17]. When diverse viruses seek to manipulate p53, they thereby impact a number of host defenses. Here, we suggest that increasing p53 levels using SGT-53 will likewise have far-reaching effects that are likely to curtail viral infections more effectively than selecting an individual therapeutic target downstream of p53. We believe that SGT-53 warrants assessment as an antiviral therapeutic in patients with COVID-19. Airway epithelial cells, AMs, which are a first line of defense against respiratory pathogens, and pDCs, which are a major source of IFN-1, are important cells in a coronavirus infection. All three express TfR (CD71), so all of these cell types should be targeted with SGT-53. Perhaps the most feasible way to test our hypothesis would be in hospitalized patients with COVID-19 who are not receiving mechanical ventilation but who are administered intravenous SGT-53 or placebo in addition to standard care. The primary endpoint of such a trial would be the fraction of patients who are on mechanical ventilation or dead by a prespecified time. The results of an analogous trial have been published, with tocilizumab, a monoclonal antibody against the IL-6 receptor, being added to standard care for patients with COVID-19 [99]. In the longer term, we speculate that it may be possible to develop SGT-53 therapy for individuals earlier in the course of the disease using self-administered intranasal SGT-53. We have evidence that intranasal administration of scL nanocomplexes in mice results in altered gene expression in lung cells. Because the p53-mediated antiviral pathway is involved in host defense against a wide variety of viruses, *TP53* gene therapy via SGT-53 has the potential of controlling viral infections against the various variants of SARS-CoV-2 that have emerged or will emerge in the course of the COVID-19 pandemic. Perhaps even more significantly, the general nature of the antiviral effect of p53 overexpression suggests that SGT-53 would be effective against the next emergence of a heretofore unknown virus, whether it be another coronavirus or a member of another virus family. In this way, the antiviral approach proposed here is distinguished from one based on vaccines in which prophylactic efficacy is constrained to the specific viral infections for which the vaccine was designed.

## Figures and Tables

**Figure 1 viruses-14-00739-f001:**
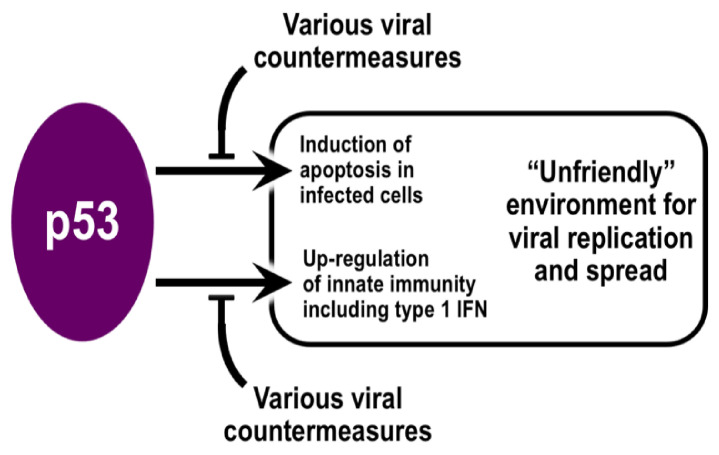
Viruses and their hosts do battle around p53. Important components of the cellular antiviral response involve p53, which is better known as a tumor suppressor. Primary among the multiple pathways regulated by p53 are induction of apoptosis of virally infected cells and promotion of type 1 interferon (IFN) response. Both of these p53-mediated processes contribute to creating a more “unfriendly” environment for viral replication and spread. For their part, viruses (both DNA and RNA) have adapted during evolution to the host antiviral responses and collectively possess an impressive repertoire of means to interfere with p53-dependent host defense mechanisms. These include encoding proteins that associate with and affect the functionality of p53, post-translational modifications of p53, and reducing p53 levels via its proteolytic destruction.

**Figure 2 viruses-14-00739-f002:**
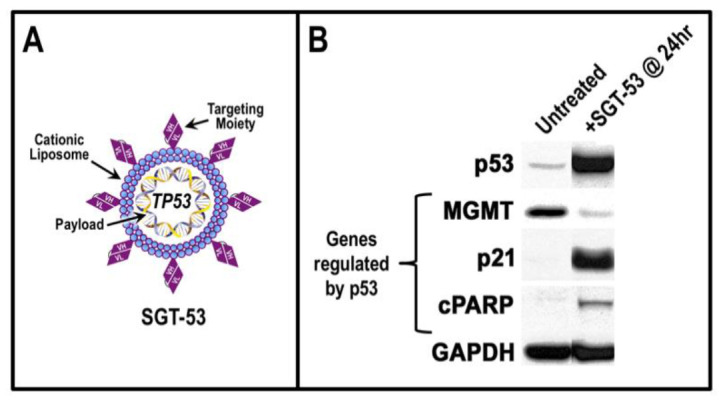
Expression of active p53 in intracranial glioblastoma after a systemic administration of SGT-53. (**A**) Schematic representation of SGT-53 consisting of a cationic liposome decorated with a targeting moiety (a single-chain antibody fragment recognizing the transferrin receptor) that carries a plasmid expression vector with the human *TP53* gene encoding wild-type p53. (**B**) Western blots showing expression of p53 in intracranial glioblastoma tumors 24 h after intravenous administration of SGT-53 (see Kim et al. [79] for experimental methods). Also shown are products of three genes regulated by p53: methylguanine-DNA methyltransferase (MGMT); cyclin-dependent kinase inhibitor (p21); and poly(ADP-ribosyl)transferase (cPARP). Glyceraldehyde-3-phosphate dehydrogenase (GAPDH) is a “house-keeping” protein.

**Figure 3 viruses-14-00739-f003:**
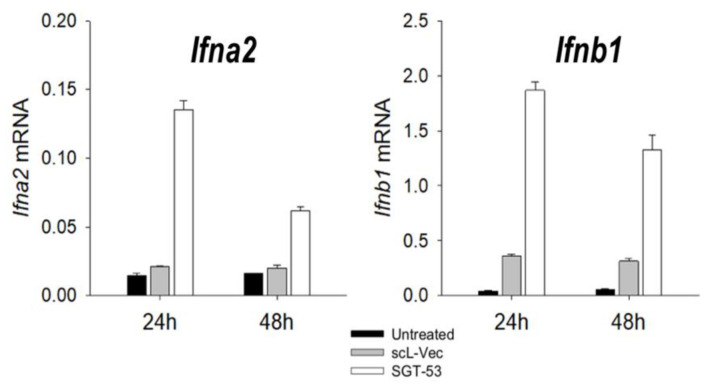
Expression of IFN-1 genes are upregulated by treatment of GBM cells with SGT-53. Mouse glioblastoma GL261 cells were transfected with SGT-53 or scL-Vec, and expressions of Ifna2 and Ifnb1 were assessed by RT-qPCR using RNA isolated at 24 or 48 h. See Kim et al. [83] for experimental methods. scL-Vec is a nanocomplex analogous to SGT-53 but carries a plasmid vector lacking the *TP53* gene.

**Figure 4 viruses-14-00739-f004:**
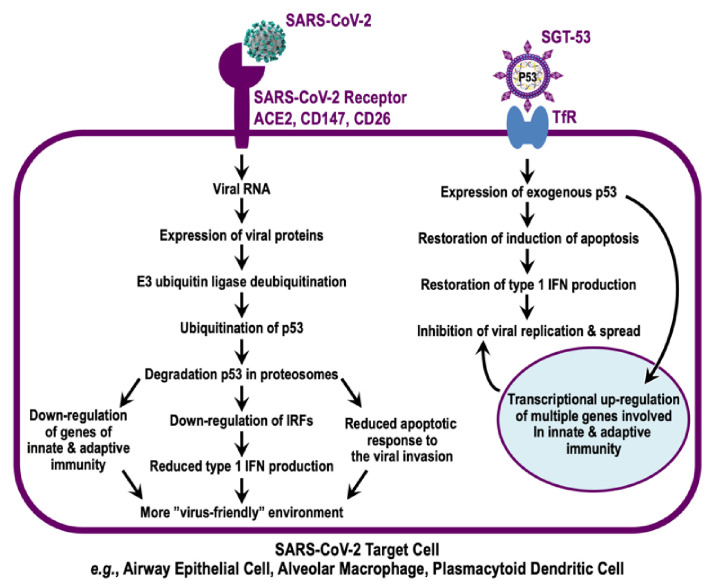
*TP53* gene therapy via SGT-53 to counter the attempt of SARS-CoV-2 to reduce p53 levels. Airway epithelial cells are the initial site of SARS-CoV-2 infection and replication. These cells express both viral receptors and TfRs. The former allows infection by SARS-CoV-2, and the latter affords the ability of the cells to take up SGT-53 for expression of exogenous p53. AMs are prominent sentinel cells in innate immunity. AMs and pDCs, which produce massive amounts of IFN-1, also possess both viral receptors and TfRs. Upon infection, viral RNA gives rise to viral proteins, notably viral nsp3, capable of deubiquitinating cellular E3 ubiquitin ligases (e.g., MDM2 and RCHY1). This leads to ubiquitination of p53 and its destruction in proteasomes. Lower p53 levels result in a more “viral-friendly” environment via downregulation of genes of innate and adaptive immunity, reduced IFN-1 production, and reduced apoptosis. Treatment with SGT-53 results in the expression of exogenous p53, which regulates a number of genes involved in innate and adaptive immunity and restores IFN production and apoptosis to inhibit viral replication and spread to other cells.

## Data Availability

Not applicable.

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
