# Peer review of "TP53* Gene Therapy as a Potential Treatment for Patients with COVID-19"

_viruses, 2022, doi:10.3390/v14040739_

Round 1

Reviewer 1 Report

The authors have proposed a novel treatment, SGT-53 gene therapy for patients suffering from COVID-19. The therapy is currently in clinical trial for pancreatic cancer and has also shown great promise as an antiviral drug.

Suggestions for the authors:

  1. The title can be changed to 'TP53 Gene Therapy as a Potential Treatment for Covid-19 Patients'
  2. Line 110 is not clear. Is the sentence trying to say that BBC3 and BAX genes are downregulated in Hepatitis B as p53 DNA binding sites are blocked?

3.Discussion needs these following elaborations:

 a. What in vitro and in vivo experiments need to be performed with SGT-53 to test its potency as a treatment against COVID-19, since no tests have been done with this drug so far.

b. What can be some of the limitations or side effects of using SGT-53 as an antiviral drug? Can overexpression of p53 cause apoptosis of healthy cells?

Line 376 human spelling needs correction.

Author Response

Response to Reviewer 1`

  1. The title was modified.  We chose to use "Patients with Covid-19" rather than "Covid-19 patients".  In my time at NCI, I learned that patients would prefer NOT to have their disease be used as an adjective in describing them.  They are a person with a disease.  Nonetheless the new title is a bit shorter so thanks to the reviewer for the suggestion.  The change to “patients with Covid-19” was made throughout the paper.
  2. The formerly confusing description of BBC3 and BAX has been reworded to make the intent more clear (hopefully).  The new wording is now in lines 112-115 of the revised manuscript.
  3. a. A comment on future experiments to provide proof-of-principle for TP53 gene therapy against SARS-CoV2 (lines 396-399 of revised manuscript); b. Two comments have also been added (lines 405-410 of revised manuscript) that are related to potential side effects of SGT-53 administration.  Firstly, we note that patients with cancers treated with multiple (>50 in some cases) doses of SGT-53 have not experienced side effects related to triggering of apoptosis in normal cells.  Secondly, we added a comment on the interesting observation that elephants have ~20 copies (40 alleles) of TP53 without their normal cells undergoing apoptosis although elephant cells are more prone to apoptosis triggered by DNA damage.  Although this may appear to be an unrelated observation, it does address the issue raised by the reviewer re. the impact of additional copies of TP53 in normal cells.

The misspelling pointed out by the reviewer was corrected (along with a few others discovered upon re-reading the paper.

Note that to address the reviewers’ comments, a few additional references were added that required renumbering of those in the earlier version.

The authors thank this reviewer and feel the paper has been improved by the process.

Reviewer 2 Report

You present all the background information and rationale for building your case of p53 involvement for COVID treatment.

What is missing though is the virus replication info; it is well established that CovSARS replicate in the resp epithelia but you seem to ignore this fact in the manuscript.

Your hypothesis would be valid if u could target SGT onto resp epithelia and prevent further viral replication and spread.

Author Response

Response to Reviewer 2`

As noted by the reviewer, we had not spoken specifically about airway epithelial cells per se but had mentioined both aveolar macrophages and plasmacytoid dendritic cells because of their involvement in innate immunity including interferon production.  We modified the manuscript to address this omission noted by the reviewer.  The new text is found in lines 312-326 of the revised manuscript.  We added references for the finding that certain inhaled viruses (New World arenaviruses) actually use the TfR as their receptor for entering airway epithelial cells.  It appears that as depicted in Figure 4, that airway epithelial cells, alveolar macrophages and plasmacytoid dendritic cells all possess both receptors for SARS-Cov-2 and transferrin.  This means that the targeting moiety on SGT-53 (TfRscFv) should allow the nanocomplex for TP53 gene therapy to enter those lung cells involved in viral replication and spread.  A modification of Figure 4 and its legend have been made to address this reviewer’s comment.

Note that to address the reviewers’ comments, a few additional references were added that required renumbering of those in the earlier version.

The authors thank this reviewer and feel the paper has been improved by the process.

This manuscript is a resubmission of an earlier submission. The following is a list of the peer review reports and author responses from that submission.

Round 1

Reviewer 1 Report

Authors wrote a perspective concerning the use of their novel investigational agent SGT-53 as a therapy for infection by SARS-CoV-2, the pathogen responsible for the Covid-19 pandemic.

They indeed provide data from literature for the strategies that various viruses develop to overcome the normal function of p53 protein as an important host response to viral infections.

p53, as a pleiotropic transcription factor, regulates various cellular pathways such as cell cycle, differentiation, apoptosis by intervening in the expression of numerous genes.    .

As for the viruses, some of them need active p53 to proliferate, while others downregulate the p53 to be active, as is the case provided by the authors for the SARS-COV-2.

SARS-COV-2 is an RNA virus, that gave the still going COVID-19 pandemic and although too many research groups deal with its mechanism of action leading to the pathophysiology, it is still many areas of its behaviors that are not clearly understood.

SARS-CoV-2 also gain mutations over time, resulting in genetic variation in the population

In oncology, many treatments such as chemotherapy impose serious adverse side effects

on patients and targeted gene therapy raised as an alternative. However, although there are already numerous clinical trials of p53 gene therapy, there are still safety concerns that limited its use in clinical application

Thus, in case of COVID 19, that affected the global population, I think there are not efficient data for what the authors suggest for “repurposing the SGT-53 as a therapy for the Covid-19 pandemic”